# Prenatal fortified balanced energy-protein supplementation and birth outcomes in rural Burkina Faso: A randomized controlled efficacy trial

**Brenda de Kok**[1☯], **Laeticia Celine Toe**[1,2☯], **Giles Hanley-Cook**[1☯], **Alemayehu Argaw**[1,3], **Moctar Ouédraogo**[4], **Anderson Compaoré**[4], **Katrien Vanslambrouck**[1], **Trenton Dailey-Chwalibóg**[1], **Rasmané Ganaba**[4], **Patrick Kolsteren**[1], **Lieven Huybregts**[1,5], **Carl Lachat**[1]*

**1** Department of Food Technology, Safety and Health, Faculty of Bioscience Engineering, Ghent University, Ghent, Belgium, **2** Unité Nutrition et Maladies Métaboliques, Institut de Recherche en Sciences de la Santé (IRSS), Bobo-Dioulasso, Burkina Faso, **3** Department of Population and Family Health, Institute of Health, Jimma University, Jimma, Ethiopia, **4** AFRICSanté, Bobo-Dioulasso, Burkina Faso, **5** Poverty, Health, and Nutrition Division, International Food Policy Research Institute (IFPRI), Washington, DC, United States of America

☯ These authors contributed equally to this work.
* Carl.Lachat@UGent.be

**Data Availability Statement:** The informed consent form does not allow sharing of personal data outside the research team. Request to access

## Abstract

### Background

Providing balanced energy–protein (BEP) supplements is a promising intervention to improve birth outcomes in low- and middle-income countries (LMICs); however, evidence is limited. We aimed to assess the efficacy of fortified BEP supplementation during pregnancy to improve birth outcomes, as compared to iron–folic acid (IFA) tablets, the standard of care.

### Methods and findings

We conducted an individually randomized controlled efficacy trial (MIcronutriments pour la SAnté de la Mère et de l'Enfant [MISAME]-III) in 6 health center catchment areas in rural Burkina Faso. Pregnant women, aged 15 to 40 years with gestational age (GA) <21 completed weeks, were randomly assigned to receive either fortified BEP supplements and IFA (intervention) or IFA (control). Supplements were provided during home visits, and intake was supervised on a daily basis by trained village-based project workers. The primary outcome was prevalence of small-for-gestational age (SGA) and secondary outcomes included large-for-gestational age (LGA), low birth weight (LBW), preterm birth (PTB), gestational duration, birth weight, birth length, Rohrer's ponderal index, head circumference, thoracic circumference, arm circumference, fetal loss, and stillbirth. Statistical analyses followed the intention-to-treat (ITT) principle. From October 2019 to December 2020, 1,897 pregnant women were randomized (960 control and 937 intervention). The last child was born in August 2021, and birth anthropometry was analyzed from 1,708 pregnancies (872 control

data need to be directed to the ethics committee of Ghent University Hospital through ethisch. comite@uzgent.be. Supporting study documents, including the study protocol and questionnaires, are publicly available on the study's website: https://misame3.ugent.be.

**Funding:** This work was supported by the Bill & Melinda Gates Foundation (Grant number: OPP1175213; https://www.gatesfoundation.org/) awarded to PK and all research team members. Under the grant conditions of the Foundation, a Creative Commons Attribution 4.0 Generic License has already been assigned to the Author Accepted Manuscript version that might arise from this submission. The funder had no role in study design, data collection and analysis, decision to publish, or preparation of the manuscript.

**Competing interests:** The authors have declared that no competing interests exist.

**Abbreviations:** ANC, antenatal care; BEP, balanced energy–protein; BMI, body mass index; CI, confidence interval; CONSORT, Consolidated Standards of Reporting Trials; DSMB, Data and Safety Monitoring Board; GA, gestational age; Hb, hemoglobin; IFA, iron–folic acid; ITT, intention-to-treat; LBW, low birth weight; LGA, large-for-gestational age; LMIC, low- and middle-income country; LNS, lipid-based nutrient supplement; MISAME, MIcronutriments pour la SAnté de la Mère et de l'Enfant; MMN, multiple micronutrients; MUAC, mid-upper arm circumference; pp, percentage points; PTB, preterm birth; RCT, randomized controlled trial; RE, retinol equivalent; SAE, serious adverse event; SD, standard deviation; SGA, small-for-gestational age; WHO, World Health Organization.

and 836 intervention). A total of 22 women were lost to follow-up in the control group and 27 women in the intervention group. BEP supplementation led to a mean 3.1 percentage points (pp) reduction in SGA with a 95% confidence interval (CI) of −7.39 to 1.16 ($P = 0.151$), indicating a wide range of plausible true treatment efficacy. Adjusting for prognostic factors of SGA, and conducting complete cases (1,659/1,708, 97%) and per-protocol analysis among women with an observed BEP adherence ≥75% (1,481/1,708, 87%), did not change the results. The intervention significantly improved the duration of gestation (+0.20 weeks, 95% CI 0.05 to 0.36, $P = 0.010$), birth weight (50.1 g, 8.11 to 92.0, $P = 0.019$), birth length (0.20 cm, 0.01 to 0.40, $P = 0.044$), thoracic circumference (0.20 cm, 0.04 to 0.37, $P = 0.016$), arm circumference (0.86 mm, 0.11 to 1.62, $P = 0.025$), and decreased LBW prevalence (−3.95 pp, −6.83 to −1.06, $P = 0.007$) as secondary outcomes measures. No differences in serious adverse events [SAEs; fetal loss (21 control and 26 intervention) and stillbirth (16 control and 17 intervention)] between the study groups were found. Key limitations are the nonblinded administration of supplements and the lack of information on other prognostic factors (e.g., infection, inflammation, stress, and physical activity) to determine to which extent these might have influenced the effect on nutrient availability and birth outcomes.

## Conclusions

The MISAME-III trial did not provide evidence that fortified BEP supplementation is efficacious in reducing SGA prevalence. However, the intervention had a small positive effect on other birth outcomes. Additional maternal and biochemical outcomes need to be investigated to provide further evidence on the overall clinical relevance of BEP supplementation.

## Trial registration

ClinicalTrials.gov NCT03533712.

---

Author summary

### Why was this study done?

- Previous studies showed that balanced energy–protein (BEP) supplementation led to a reduction in small-for-gestational age (SGA) and low birth weight (LBW) babies and stillbirth, and increased birth weight.

- Conclusions on the impact of BEP should be interpreted with caution due to the large heterogeneity in supplement types administered and their timing, study populations, and quality of the study designs.

- There is a critical need for high-quality randomized controlled trials (RCTs) with adequate sample sizes, to assess the efficacy of BEP during pregnancy to support fetal growth and improve birth outcomes.

## What did the researchers do and find?

- We performed an individually randomized controlled efficacy trial (MIcronutriments pour la SAnté de la Mère et de l'Enfant [MISAME]-III) to assess the effect of fortified BEP supplementation during pregnancy on birth outcomes in rural Burkina Faso.

- Early enrollment (gestational age [GA] approximately 11 weeks at enrollment), high adherence rates (observed daily BEP intake approximately 83%), and reliable measurements of birth outcomes (at latest 12 hours after delivery) add to the quality of the study and robustness of the findings.

- The trial did not provide evidence that BEP is efficacious in reducing SGA prevalence, but gestational duration was slightly longer, prevalence of LBW babies lower, and birth weight, birth length, and thoracic and arm circumference higher.

## What do these findings mean?

- Our findings are consistent with previous research showing that prenatal BEP led to a reduction in LBW babies and increased birth weight. However, we did not observe a statically significant effect on SGA, in comparison to earlier evidence.

- Future research in MISAME-III will assess additional maternal and child biochemical parameters to provide further insights into the clinical relevance of BEP supplementation.

## Introduction

Improving fetal and newborn health remains a global challenge with an estimated 20 million infants a year born low birth weight (LBW; <2,500 g) [1], 14 million born preterm [2], and 23 million born small-for-gestational age (SGA; <10th percentile of a reference population) [3]. Preterm birth (PTB; born <37 completed weeks of gestation) and SGA babies have an increased mortality risk in the first year of life [4] and a higher likelihood to develop noncommunicable diseases in adulthood [5]. Undernutrition during pregnancy is an important risk factor for intrauterine growth restriction, poor fetal development, and suboptimal newborn health [6]. In many low- and middle-income countries (LMICs), inadequate dietary intakes and increased physiological demands result in maternal macro- and micronutrient deficiencies [7].

At present, the World Health Organization (WHO) recommends daily oral iron–folic acid (IFA) supplementation as part of routine antenatal care (ANC) to mitigate the risks for neural tube defects, LBW, maternal anemia, and iron deficiency [8]. Prenatal multiple micronutrient (MMN) and balanced energy–protein (BEP; <25% of total kcal content from protein) supplementation are alternative strategies proposed to address maternal nutrient deficiencies and to improve subsequent birth outcomes [9].

Previous evidence indicates that prenatal BEP supplementation led to a reduction in the risks of SGA, LBW, stillbirth, and increased birth weight, in particular among malnourished women [10,11]. Based on this evidence, WHO recommends prenatal BEP supplements in

populations with a prevalence of >20% underweight pregnant women (body mass index [BMI] <18.5 kg/m$^2$) [8].

More recent studies, however, show mixed results on the effectiveness of nutritional supplements. The Women First trial, a multicountry randomized controlled trial (RCT), indicated a positive effect of small-quantity lipid-based nutrient supplement (LNS; 118 kcal) before conception or late in the first trimester on mean birth size and SGA [12]. However, an intervention in Niger, which offered medium-quantity LNS (237 kcal), did not report an impact on birth weight [13]. Similarly, an RCT in Malawi that provided a large quantity LNS (920 kcal) to malnourished pregnant women found no effect on birth weight and length [14].

The huge heterogeneity in the type of supplements (i.e., either MMN fortified, ready to use, lipid based, or not), study designs, inclusion criteria, and comparison or control groups make it difficult to draw firm conclusions on BEP supplementation. Specifically designed efficacy studies are therefore needed to assess the importance of investing in prenatal BEP supplements to support fetal growth and improve birth outcomes.

The previous efficacy study "MIcronutriments pour la SAnté de la Mère et de l'Enfant" (MISAME)-II assessed the effect of a prenatal fortified BEP, with plant-based protein, as compared to a MMN tablet on birth size in rural Burkina Faso. Although a positive effect was observed for birth length (0.6 to 6.7 mm increase), the intervention did not reduce SGA prevalence or preterm delivery [15]. The use of an active comparator group with MMN could have potentially led to a masking effect of BEP on SGA, as it has been reported that MMN supplementation decreased LBW, SGA, and PTB [16]. As many countries provide IFA as the standard of care, the effect of BEP should be assessed relative to this standard. In the MISAME-III study, we assess the effect of a daily prenatal fortified BEP supplement—a large quantity LNS with milk protein—and IFA tablet on SGA prevalence and other birth outcomes, as compared to a daily IFA.

## Methods

Our research was reported using the Consolidated Standards of Reporting Trials (CONSORT) 2010 checklist (S1 CONSORT checklist) [17].

### Study design and participants

The MISAME-III protocol was published previously [18]. In brief, the study was a community-based, nonblinded individually randomized 2 × 2 factorial RCT, with directly observed daily supplement intake, conducted in the Houndé health district situated in the Hauts-Bassins region of Burkina Faso. The present manuscript details the primary and secondary birth outcomes only. The maternal and postnatal study outcomes will be reported separately. The study protocol was approved by the ethics committee of Ghent University Hospital in Belgium (B670201734334) and Centre Muraz in Burkina Faso (N˚2018–22/MS/SG/CM/CEI).

Women aged between 15 and 40 years and living in the study catchment villages were identified through a census in the study area ($n$ = 10,165). A network of 142 trained village-based project workers visited all eligible women at their homes every 5 weeks to identify pregnancy early, by screening for self-reported amenorrhea. Potential cases were referred to the health center for a urinary pregnancy test. Once gestation was confirmed, the MISAME-III study purpose and procedures were explained in the local languages Mooré, Dioula, or Bwamu. Study eligibility criteria were (i) pregnancy confirmed by a urinary pregnancy test and ultrasound examination; and (ii) written informed consent. Exclusion criteria were (i) gestational age (GA) ≥21 completed weeks; (ii) women who planned to leave the area during their pregnancy or deliver outside the study area; and (iii) women allergic to peanuts. Study inclusion

ran from October 30, 2019 to December 12, 2020, and the final child was born on August 7, 2021.

The climate of the study setting is Sudano-Sahelian, with one dry season conventionally running from September to October to April. Malaria transmission is perennial, with seasonal variations. Regional health statistics from the 6 healthcare centers showed that 1.8% of adults suffered from hookworm or another parasitic infection and 0.9% from a sexually transmitted disease in 2021. The prevalence of pregnant women that suffered from a HIV infection was estimated to be 0.7% [19].

## Randomization and masking

We randomly allocated women to the prenatal control or intervention group. A stratified permuted block randomization schedule was used to allocate women to the study groups. These blocks were generated per health center in blocks of 8 (4 control and 4 intervention) before the start of the study using Stata V.15.1 (Stata, College Station, Texas, United States of America) by a research analyst who was not involved in the study (FB). The allocation was coded with the letters A for the prenatal control and B for the prenatal intervention group and concealed in sequentially numbered sealed opaque envelopes by study employees, not in direct contact with participants. The study midwives, who enrolled the participants, assigned the women to the study groups by drawing a next sealed envelope with the letter code. Postrandomization, we excluded women without a confirmed pregnancy using the ultrasound examination, women with GA $\geq$21 completed weeks, and multifetal pregnancies [20].

It was not possible to blind the supplement allocation from study participants and trained village-based project workers because the products are readily identifiable. Outcome assessors (study physician, midwives, and field supervisors) were different from study collaborators (trained village-based project workers) who distributed the study supplements. However, given the nonblinded nature of the study, outcome assessors could have been aware of the study group allocation by asking the mother. Researchers who analyzed the data were not blinded.

## Procedures

Women in the intervention group received a daily BEP supplement and IFA tablet for the duration of their pregnancy. In a formative study, the most preferred and suitable fortified BEP supplement was selected for administration in the MISAME-III efficacy trial [21,22]. The BEP supplement is an LNS in the form of an energy-dense peanut paste fortified with MMNs. The product is ready to consume, does not require a cold chain, and is highly stable with a long shelf life. On average, the 72g fortified BEP provided 393 kcal and consisted of 36% lipids, 20% protein, and 32% carbohydrates. Protein came from soy (61%), milk (25%), and peanut (15%). Furthermore, the MMN content covered at least the daily estimated average requirements of micronutrients for pregnant women, except for calcium, phosphorous, and magnesium, which were lower [23]. The complete nutritional composition of the fortified BEP is provided in Table 1 [24]. Women in the control group received daily only an IFA tablet (65 mg iron [form: $FeH_2O_5S$] and 400 μg folic acid [form: $C_{19}H_{19}N_7O_6$]; Sidhaant Life Sciences, Delhi, India), in accordance with the standard of care in Burkina Faso.

Both supplements were delivered on a daily basis and, to the extent possible, consumed under supervision by our trained village-based project workers during home visits. When women had a short and scheduled absence of home, supplements were given to the women in advance, and intake was considered nonobserved for the respective days. The trained village-based project workers also encouraged pregnant women to attend at least 4 ANC

**Table 1. Nutritional values of the BEP supplement for pregnant women[a].**

| | Mean for 72 g (serving size) |
|---|---|
| Total energy (kcal) | 393 |
| Lipids (g) | 26 |
| Linoleic acid (g) | 3.9 |
| α-Linoleic acid (g) | 1.3 |
| Proteins (g) | 14.5 |
| Carbohydrates (g) | 23.3 |
| Calcium (mg) | 500 |
| Copper (mg) | 1.3 |
| Phosphorus (mg) | 418 |
| Iodine (μg) | 250 |
| Iron (mg) | 22 |
| Selenium (μg) | 65 |
| Manganese (mg) | 2.1 |
| Magnesium (mg) | 73 |
| Potassium (mg) | 562 |
| Zinc (mg) | 15 |
| Vitamin A (μg RE)[b] | 770 |
| Thiamin (mg) | 1.4 |
| Riboflavin (mg) | 1.4 |
| Niacin (mg) | 15 |
| Vitamin B5 (mg) | 7 |
| Vitamin B6 (mg) | 1.9 |
| Folic acid (μg) | 400 |
| Vitamin B12 (mg) | 2.6 |
| Vitamin C (mg) | 100 |
| Vitamin D (μg cholecalciferol)[c] | 15 |
| Vitamin E (mg α-tocopherol)[d] | 18 |
| Vitamin K (μg) | 72 |

[a]Ingredients: vegetable oils (rapeseed, palm, and soy in varying proportions), defatted soy flour, skimmed milk powder, peanuts, sugar, maltodextrin, soy protein isolate, vitamin and mineral complex, and stabilizer (fully hydrogenated vegetable fat and mono- and diglycerides).

[b]1 μg vitamin A RE = 3.333 IU vitamin A.

[c]1 μg cholecalciferol = 40 IU vitamin D.

[d]1 mg α-tocopherol = 2.22 IU vitamin E.

BEP, balanced energy–protein; IU, international unit, RE, retinol equivalent.

consultations. Study participants were designated as lost to follow-up if they moved from the study area, withdrew their participation, or if they could not be reached for more than 3 months.

At enrollment (i.e., first ANC visit), pregnancy antecedents were collected and maternal height, weight, mid-upper arm circumference (MUAC), and hemoglobin (Hb) concentration were measured. Maternal height was measured to the nearest 1 cm with a ShorrBoard Infant/ Child/Adult (Weigh and Measure, Olney, Maryland, USA) and weight to the nearest 100 g with a Seca 876 scale (Seca, Hanover, Maryland, USA); the accuracy of the scales was verified on a weekly basis. Maternal MUAC was measured to the nearest 1 mm with a Seca 212 measuring tape. Hb concentration was assessed again between 30 and 34 weeks of gestation (i.e.,

third ANC visit) using a HemoCue Hb 201+ (HemoCue, Ängelholm, Sweden); a calibration check was done weekly. Furthermore, a comprehensive socioeconomic and demographic questionnaire was administered at enrollment [18].

During each subsequent ANC visit, the study midwives measured all anthropometrics and screened for potential adverse events by checking blood pressure, urine protein, body temperature, edema, and fetal activity. Following Burkinabè guidelines, enrolled women received preventative malaria prophylaxis (3 oral doses of sulfadoxine–pyrimethamine) at the relevant ANC visits.

Within 14 days of enrollment date, a woman's pregnancy was confirmed by the study physician using a portable ultrasound (SonoSite M-Turbo, FUJIFILM SonoSite, Bothell, Washington, USA). GA was estimated by measuring crown-rump length (7 to 13 weeks) or by calculating the mean of 3 to 4 measurements: biparietal diameter, head circumference, abdominal circumference, and femur length (12 to 26 weeks) [25]. In addition to the ultrasound, the physician performed maternal subscapular and tricipital skinfold measurements in triplicate using a Harpenden caliper.

At birth, anthropometry of all neonates was assessed in duplicate within the first 72 hours by study midwives (in practice, all were within 12 hours) at the health center. Newborn length was measured to the nearest 1 mm with a Seca 416 Infantometer, whereas birth weight was measured to the nearest 10 g with a Seca 384 scale. Newborn head circumference, thoracic circumference, and MUAC were measured to the nearest 1 mm with a Seca 212 measuring tape. If there was a large discrepancy between measures (e.g., >10 mm for birth length and >200 g for birth weight), a third measurement was taken. The average of the 2 closest measures were used for analyses. The accuracy and precision of anthropometric measurements were established regularly through standardization sessions organized by an expert in anthropometry [26].

MISAME-III data were collected using SurveySolutions (version 21.5) on tablets by the study physician and midwives and were transferred to a central server at Ghent University on a weekly basis. Questionnaire assignments were sent to the field team once a week including preloaded data collected at the previous ANC visit. We programmed generic validation codes to avoid the entry of implausible values and improve the quality of data collection in the field. Additionally, data quality checks and missing or inconsistent data were sent back to the field for revision every 2 weeks. The quality of ultrasound images and estimation of GA was checked for 10% of the examinations on a regular basis by an external gynecologist, using a quality checklist and scoring sheet. The MISAME-III trained village-based project workers collected data on the supplement adherence in both prenatal study groups using smartphones with computer-assisted person interviewing programmed in CSPro (version 7.3.1) on a daily basis. Six field supervisors performed monthly quality checks by verifying a trained village-based project worker's work, at random, using a Lot Quality Assurance Sampling system [27].

All field staff received extensive training on all standard operating procedures (including Good Clinical Practices) and data collection tools before the start of the trial, with a dry run period of ±3 months for testing and evaluation in the field. The MISAME-III data collection forms are publicly available [28].

## Outcomes

The primary study outcome was the prevalence of SGA, defined as the proportion of newborns with a birthweight below the 10th percentile of the International Fetal and Newborn Growth Consortium for the 21st Century [INTERGROWTH-21st] newborn size standards for a given GA at delivery [29].

The secondary outcomes of the prenatal BEP intervention were prevalence of large-for-gestational age (LGA; >90th percentile of INTERGROWTH-21st reference), LBW and PTB, gestation duration (weeks), birth weight (g), birth length (cm), Rohrer's ponderal index at birth [weight/length$^3$ (g/cm$^3$) × 1,000], head circumference (cm), thoracic circumference (cm), arm circumference (mm), fetal loss (<28 weeks of gestation), and stillbirth (died ≥28 weeks of gestation, before or during birth). Fetal loss is further categorized in (i) <22 weeks of gestation; (ii) between ≥22 weeks and <28 weeks of gestation, according to the "Maternal BEP studies Harmonization Initiative". Birth length and Rohrer's ponderal index were measured to distinguish between short and thin newborns.

To assess safety and serious adverse events (SAEs), all field staff was trained to recognize pregnancy related health issues to actively refer participants to the health center. All SAEs (i.e., miscarriage, stillbirth, and maternal death) were recorded on a case-by-case basis, and verbal autopsies were conducted for infant and/or maternal deaths that occurred outside a health center.

## Statistical analysis

All analyses were documented in the MISAME-III statistical analysis plan prior to analysis, which was validated on October 24, 2019 and published online on November 3, 2020 [28].

We calculated a sample size of 652 pregnant women per prenatal study group (total 1,304 participants) to detect a decrease in SGA of 7 percentage points (pp) between groups, with a power of 80% and a 2-sided significance level (i.e., type I error) of 5%, assuming a SGA prevalence of 32% (estimated from Huybregts and colleagues [15]). In MISAME-I [30] and MISAME-II [15], an approximately 26% loss of information occurred, due to a combination of abortions, miscarriages, stillbirths, multifetal pregnancies, out-migrations, maternal deaths, and incomplete data. Hence, the sample size was increased to 888 pregnant women per prenatal study group to accommodate for these potential losses (total 1,776 participants).

Only singleton pregnancies were included in the analysis, as anthropometric measures and fetal loss at birth in multifetal pregnancies are often not primarily nutrition related. The primary analysis followed the intention-to-treat (ITT) principle. Therefore, we conducted multiple imputation by chained equations of missing outcome measures at birth under the "missing at random" assumption. A total of 50 imputations of missing values were done for the lost to follow-up cases to estimate the regression coefficients using the predictors maternal height, BMI, MUAC, Hb, age, GA and primiparity at baseline, and month of inclusion.

Descriptive data are presented as percentages or means ± standard deviation (SD). Unadjusted and adjusted group differences were estimated by fitting linear regression models for continuous outcomes to estimate the mean group difference. For binary outcomes, linear probability models with a robust variance estimator were used to estimate risk difference in pp. All models contained health center and randomization block as fixed effect to account for any possible clustering by the study design. The adjusted models contained a priori defined known prognostic factors of study outcomes at birth, including maternal height (cm), BMI (kg/m$^2$), MUAC (mm), Hb (g/dl), age (years) and GA at inclusion (weeks), and primiparity. Due to balanced baseline characteristics across prenatal study groups (i.e., < |2.5| pp difference), no other sociodemographic variables were adjusted for in sensitivity analyses.

We conducted the following sensitivity analyses to assess the robustness of the primary findings: (i) complete case analysis (i.e., excluding women lost to follow-up); and (ii) per-protocol analysis restricting the intervention sample to women with BEP adherence of ≥75%. The strict adherence rate was calculated by dividing the total number of BEP supplements effectively taken under direct observation of a trained village-based project worker by the

theoretical maximum number of prenatal BEP supplements, i.e., the number of days between study inclusion and delivery.

Furthermore, as an exploratory analysis, we tested an interaction term between the intervention group and pre-specified subgroups, including maternal BMI ($<18.5$ kg/m$^2$), MUAC ($<23$cm), Hb ($<11$g/dl), height ($<155$ cm), age ($<20$ years), completion of primary education, possible and probable prenatal depression (Edinburgh depression scale $\geq10$ points and $\geq13$ points), primiparity, household food insecurity (Household Food Insecurity Access Scale), newborn sex, season of delivery (lean season: June to September), and interpregnancy interval ($<18$ months). Last, we used the approach by Katz and colleagues [31] and Roberfroid and colleagues [30] to assess whether the treatment effect on birth weight and length was constant over percentiles of children's birth weight, birth length, and maternal BMI distributions. In this method, differences (and CIs) in birth outcomes between intervention and control groups are estimated as nonlinear smooth functions of the percentiles of birth weight, birth length, or maternal BMI distributions.

Statistical significance was set at $P < 0.05$ for all tests, except for exploratory interactions tests ($P < 0.10$) as specified in the statistical analysis plan. All analyses were conducted with Stata 17.1 (StataCorp).

All SAEs reported by the study physician were evaluated on a continuous basis by the principal study investigators and reported to an independent Data and Safety Monitoring Board (DSMB) when considered related to the supplement. The DSMB (established prior to the start of the efficacy trial) comprised an endocrinologist, 2 pediatricians, a gynecologist, and a medical ethicist of both Belgian and Burkinabè nationalities. Two virtual DSMB meetings were organized, at month 9 and 20 after the start of the trial, to review the study progress and discuss all documented SAEs. The MISAME-III trial was registered on ClinicalTrials.gov (identifier: NCT03533712).

## Results

From October 30, 2019 to December 12, 2020, 2,016 women were assessed for eligibility, of whom 1,897 were randomized (960 control and 937 intervention). Nine women refused to continue participation after randomization and were excluded. Subsequently, 110 women were excluded postrandomization, because pregnancy was not confirmed during the ultrasound examination. This resulted in a slight imbalance in the number of women allocated to the control and intervention groups, i.e., women who commenced IFA or IFA + BEP supplementation. Another 59 women were $\geq21$ completed weeks of gestation at inclusion and 50 women had a multifetal pregnancy and were therefore excluded from the analysis (Fig 1). The baseline characteristics of mothers included in the study (909 control and 879 intervention) are presented in Table 2. The control and intervention groups were well balanced regarding household, maternal, and pregnancy characteristics (i.e., $< |2.5|$ pp difference across groups). At baseline, 54.7% of households were food insecure, whereas 7.1% of women were underweight and 37.7% anemic.

Of the 1,788 women who were enrolled at baseline and met the inclusion criteria, 22 (2.4%) women in the control and 27 (3.1%) women in the intervention group were lost to follow-up (Fig 1). Among the 1,739 pregnancies (887 control and 852 intervention) that were followed up, there were no significant differences (all $P > 0.1$) in fetal loss (21 control and 26 intervention) or stillbirth prevalence (16 control and 17 intervention) across the groups (Table 3). No maternal deaths occurred in either the control or intervention group. The observed supplement adherence rate was 83.1% for BEP in the intervention group and 88.8% and 90.6% for IFA in the control and intervention group, respectively.

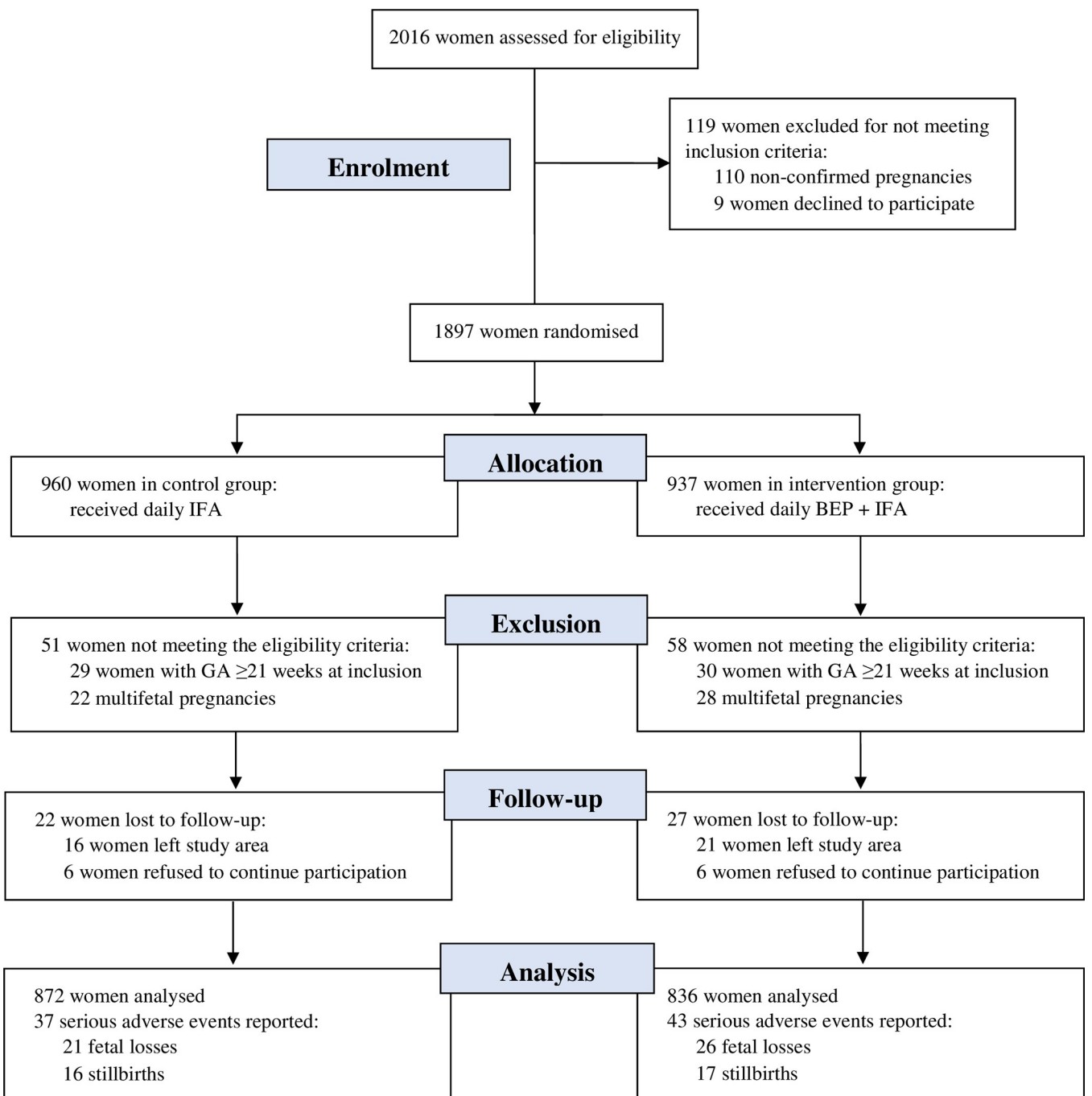

**Fig 1. CONSORT flowchart.** BEP, balanced energy–protein; CONSORT, Consolidated Standards of Reporting Trials; GA, gestational age; IFA, iron–folic acid.

BEP supplementation led to a mean 3.1 pp reduction in SGA with a 95% confidence interval (CI) of −7.39 to 1.16 (P = 0.151) (Table 4). The main finding was confirmed by adjusting the regression model for prognostic factors of SGA (−2.93 pp, −7.04 to 1.17, P = 0.161), by (un)adjusted complete cases (850 control and 809 intervention) analyses (−3.15 pp, −7.41 to 1.12, P = 0.148; S1 Table), and by (un)adjusted per protocol (850 control and 631 intervention) analyses (−3.30 pp, −7.89 to 1.29, P = 0.160; S2 Table).

**Table 2. Baseline characteristics of study participants.**

| Characteristics | Control (*n* = 909) | Intervention (*n* = 879) |
|---|---|---|
| **Health center catchment area** | | |
| Boni | 200 (22.0) | 192 (21.8) |
| Dohoun | 95 (10.5) | 97 (11.0) |
| Dougoumato II | 172 (18.9) | 154 (17.5) |
| Karaba | 93 (10.2) | 94 (10.7) |
| Kari | 167 (18.4) | 164 (18.7) |
| Koumbia | 182 (20.0) | 178 (20.3) |
| **Household level** | | |
| Wealth index, 0 to 10 points | 4.51 ± 1.74 | 4.67 ± 1.75 |
| Household food insecurity[a] | 490 (53.9) | 488 (55.5) |
| Improved primary water source[b] | 565 (62.2) | 551 (62.7) |
| Improved sanitation facility[c] | 539 (59.3) | 533 (60.6) |
| Household size | 6.19 ± 4.45 | 6.20 ± 4.21 |
| Polygamous households | 289 (31.8) | 287 (32.7) |
| **Head of household** | | |
| Age, years | 33.4 ± 9.16 | 33.8 ± 9.33 |
| Male | 906 (99.7) | 877 (99.8) |
| Completed primary education | 544 (59.8) | 519 (59.0) |
| **Maternal** | | |
| Age, years | 25.1 ± 6.20 | 25.0 ± 6.18 |
| Ethnic group | | |
| Bwaba | 521 (57.3) | 506 (57.6) |
| Mossi | 321 (35.3) | 303 (34.5) |
| Other | 67 (7.37) | 70 (7.96) |
| Religion | | |
| Muslim | 383 (42.1) | 372 (42.3) |
| Animist | 213 (23.4) | 200 (22.8) |
| Protestant | 147 (16.2) | 162 (18.4) |
| Catholic | 131 (14.4) | 115 (13.1) |
| No religion, no animist | 35 (3.85) | 30 (3.41) |
| Completed primary education | 385 (42.4) | 364 (41.4) |
| Weight, kg | 57.9 ± 8.65 | 58.4 ± 8.69 |
| Height, cm[d] | 162 ± 5.91[d] | 163 ± 6.05 |
| BMI, kg/m$^2$ | 22.0 ± 2.87 | 22.0 ± 2.87 |
| <18.5 kg/m$^2$ | 64 (7.05) | 63 (7.17) |
| MUAC, mm | 262 ± 26.8 | 262 ± 26.4 |
| Subscapular skinfold, mm | 11.9 ± 5.47 | 12.1 ± 5.58 |
| Tricipital skinfold, mm | 11.8 ± 4.76 | 12.0 ± 4.86 |
| Hb, g/dl | 11.4 ± 1.47 | 11.3 ± 1.52 |
| Anemia (Hb <11g/dl) | 334 (36.7) | 340 (38.7) |
| Severe anemia (Hb <7g/dl) | 2 (0.22) | 2 (0.23) |
| GA, weeks | 11.4 ± 4.08 | 11.5 ± 4.04 |
| Trimester of gestation | | |
| First | 574 (63.1) | 545 (62.0) |
| Second | 335 (36.9) | 334 (38.0) |
| Parity | | |
| 0 | 198 (21.8) | 203 (23.1) |

(*Continued*)

**Table 2.** (Continued)

| Characteristics | Control (*n* = 909) | Intervention (*n* = 879) |
|---|---|---|
| 1 to 2 | 326 (35.9) | 294 (33.4) |
| ≥3 | 385 (42.4) | 382 (43.5) |

Data are frequencies (%) or means ± SD.

[a]Assessed using FANTA/USAID's Household Food Insecurity Access Scale [32].

[b]Protected well, borehole, pipe, or bottled water were considered improved water sources.

[c]Flush toilet connected to local sewage or septic tank or pit latrine with slab and/or ventilation were considered improved sanitation facilities.

[d]Height of one woman with a physical disability could not be measured.

BMI, body mass index; GA, gestational age; Hb, hemoglobin; MUAC, mid-upper arm circumference; SD, standard deviation.

The (un)adjusted ITT analyses of secondary outcomes showed that the MISAME-III intervention led to significantly longer gestational duration (+0.20 weeks, 0.05 to 0.36, *P* = 0.010), birth weight (50.1 g, 8.11 to 92.0, *P* = 0.019), birth length (0.20 cm, 0.01 to 0.40, *P* = 0.044), thoracic circumference (0.20 cm, 0.04 to 0.37, *P* = 0.016), and arm circumference (0.86 mm, 0.11 to 1.62, *P* = 0.025). Moreover, prenatal BEP and IFA supplementation significantly decreased LBW prevalence (−3.95 pp, −6.83 to −1.06, *P* = 0.007), as compared to receiving IFA tablets only (Table 4). There was no significant difference between the study groups in the prevalence of LGA (0.24 pp, −0.98 to 1.46, *P* = 0.700) or PTB (−1.72 pp, −3.56 to 0.13, *P* = 0.069), thinner newborns (Rohrer's ponderal index: 0.15, −0.09 to 0.38, *P* = 0.226), or newborns with larger head circumferences (0.10 cm, −0.05 to 0.25, *P* = 0.178; Table 4).

Furthermore, (un)adjusted subgroup analyses (*P* interaction < 0.10) indicated larger intervention-related reductions in SGA prevalence among women with a baseline MUAC >23 cm, women >20 years of age, nonanemic (Hb ≥11 g/dl) women at baseline, and among female newborns (−6.73 pp, −12.6 to −0.81, *P* = 0.026; S3 Table).

Daily BEP and IFA supplementation had a stronger positive effect on birth weight and length at lower percentiles of the birth weight and length distributions, respectively (Figs 2 and 3). We did not find evidence that the treatment effect on birth weight or length was modified by maternal BMI at baseline (S1 and S2 Figs).

## Discussion

The MISAME-III trial did not provide evidence that prenatal fortified BEP supplementation was efficacious in reducing SGA prevalence. However, the intervention led to improvements

**Table 3. Fetal loss and stillbirth prevalence, by prenatal study group.**

| Definition | Control[a] (*n* = 909) | Intervention[a] (*n* = 879) | Δ[b] (95% CI) | *P* value |
|---|---|---|---|---|
| Fetal death <22 weeks of gestation | 13 (1.43) | 21 (2.39) | 1.00 (−0.25, 2.25) | 0.12 |
| Fetal death ≥22 weeks and <28 weeks of gestation | 8 (0.88) | 5 (0.57) | −0.30 (−1.10, 0.48) | 0.45 |
| Stillbirth[c] | 16 (1.76) | 17 (1.93) | 0.18 (−1.08, 1.44) | 0.78 |

[a]Values are frequencies (%).

[b]Risk differences (Δ) in pp were estimated using linear probability models with robust variance estimation, adjusted for health center and randomization block as fixed effect to account for clustering by the study design.

[c]Child died ≥28 weeks of gestation, before or during birth.

CI, confidence interval; pp, percentage points.

**Table 4. Efficacy of prenatal BEP supplementation on birth outcomes.**

| Birth characteristics | Control[a] (*n* = 872) | Intervention[a] (*n* = 836) | Unadjusted Δ[b] (95% CI) | *P* value | Adjusted Δ[b] (95% CI) | *P* value |
|---|---|---|---|---|---|---|
| SGA | 243 (27.9) | 207 (24.8) | −3.11 (−7.39, 1.16) | 0.153 | −2.93 (−7.04, 1.17) | 0.161 |
| LGA | 14 (1.55) | 15 (1.75) | 0.24 (−0.98, 1.46) | 0.700 | 0.20 (−1.01, 1.40) | 0.747 |
| LBW | 107 (12.3) | 69 (8.27) | −3.95 (−6.83, −1.06) | 0.007 | −4.07 (−6.86, −1.28) | 0.004 |
| Preterm delivery | 40 (4.65) | 25 (2.95) | −1.72 (−3.56, 0.13) | 0.069 | −1.82 (−3.67, 0.02) | 0.052 |
| GA, weeks | 39.9 ± 1.78 | 40.1 ± 1.48 | 0.20 (0.05, 0.36) | 0.010 | 0.22 (0.06, 0.37) | 0.006 |
| Birth weight, g | 2986 ± 450 | 3038 ± 427 | 50.1 (8.11, 92.0) | 0.019 | 49.7 (10.8, 88.7) | 0.012 |
| Birth length, cm | 48.2 ± 2.25 | 48.4 ± 2.13 | 0.20 (0.01, 0.40) | 0.044 | 0.20 (0.01, 0.39) | 0.037 |
| Ponderal index[c] | 26.5 ± 2.67 | 26.7 ± 2.67 | 0.15 (−0.09, 0.38) | 0.226 | 0.15 (−0.08, 0.38) | 0.208 |
| Head circumference, cm | 33.4 ± 1.64 | 33.5 ± 1.53 | 0.10 (−0.05, 0.25) | 0.178 | 0.10 (−0.04, 0.24) | 0.154 |
| Thoracic circumference, cm | 31.7 ± 1.84 | 31.9 ± 1.67 | 0.20 (0.04, 0.37) | 0.016 | 0.20 (0.05, 0.36) | 0.011 |
| Arm circumference, mm | 100 ± 8.43 | 101 ± 8.18 | 0.86 (0.11, 1.62) | 0.025 | 0.89 (0.18, 1.60) | 0.014 |

[a]Values are frequencies (%) or means ± SD.

[b]Unadjusted and adjusted group differences (Δ) were estimated by fitting linear regression models for the continuous outcomes, to estimate the mean group difference, and using linear probability models with robust variance estimation for the binary outcomes, to estimate risk difference in pp. All models contained health center and randomization block as fixed effect to account for clustering by the study design. Adjusted models additionally contained a priori set known prognostic factors of birth outcome including maternal age, primiparity, GA, height, MUAC, BMI, and Hb level at study enrollment.

[c]Ponderal index calculated as birth weight in g / (birth length in cm)$^3$ × 1,000.

BEP, balanced energy–protein; BMI, body mass index; CI, confidence interval; GA, gestational age; Hb, hemoglobin; LGA, large-for-gestational age; MUAC, mid-upper arm circumference; pp, percentage points; SD, standard deviation; SGA, small-for-gestational age.

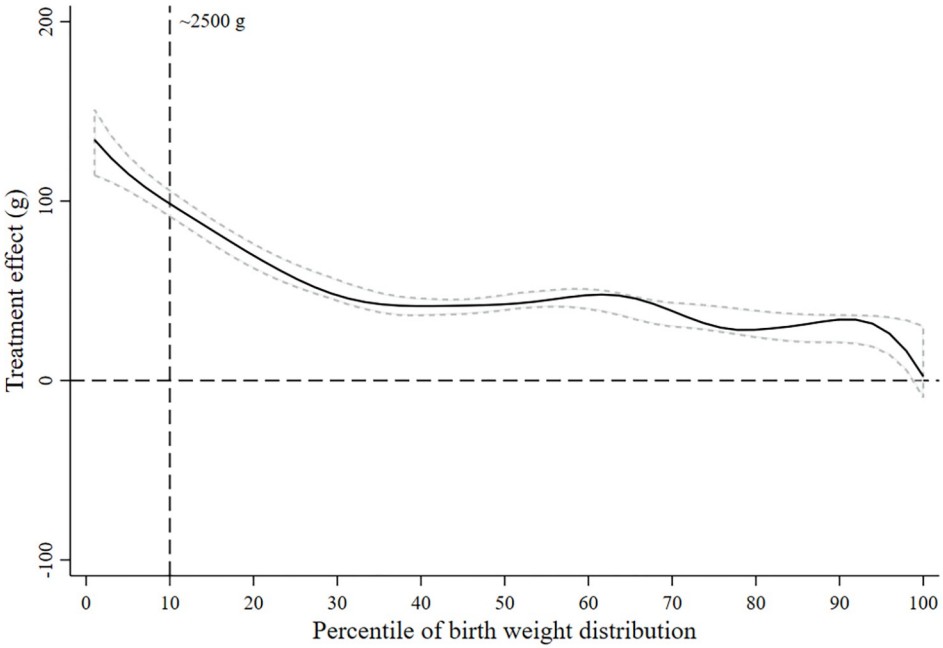

**Fig 2. Treatment efficacy on birth weight across the distribution of birth weight.** The estimated difference in birth weight between the women who received the BEP supplement and IFA (intervention) and those who received only iron and folic acid (control) is shown as a function of the percentiles of birth weights. The zero line indicates no efficacy of BEP. The positive y values indicate a higher birth weight in the intervention group, and the negative y values indicate a lower birth weight. The central solid black line represents the smoothed treatment efficacy, with upper and lower dashed 95% confidence bands, using complete cases. BEP, balanced energy–protein; IFA, iron–folic acid.

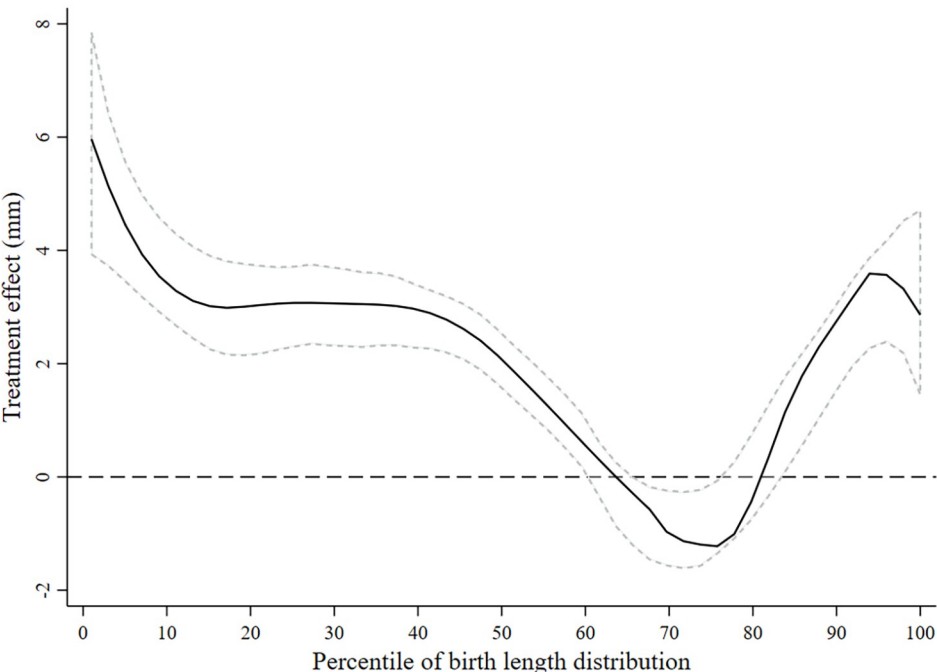

**Fig 3. Treatment efficacy on birth length across the distribution of birth length.** The estimated difference in birth length between the women who received the BEP supplement and IFA (intervention) and those who received only iron and folic acid (control) is shown as a function of the percentiles of birth lengths. The zero line indicates no efficacy of BEP. The positive y values indicate a higher birth length in the intervention group, and the negative y values indicate a lower birth length. The central solid black line represents the smoothed treatment efficacy, with upper and lower dashed 95% confidence bands, using complete cases. BEP, balanced energy–protein; IFA, iron–folic acid.

in gestational length, birth weight, birth length, thoracic and arm circumference, and decreased LBW prevalence.

This study was primarily designed to reduce the prevalence of SGA. A meta-analysis of (quasi-) RCTs concluded that prenatal (fortified) BEP supplementation resulted in a 11% to 51% (95% CI) reduction in SGA infants [10], whereas a Cochrane review of RCTs concluded that BEP led to a 10% to 31% decrease in the risk of SGA [11]. However, the supplements studied varied tremendously in terms of energy (417 to 1,017 kcal), protein (7 to 40 g), and micronutrient composition. In addition, various comparison groups and timing of supplementation were applied [33]. A direct comparison between results from these trials and our findings is therefore difficult. The previous MISAME-II trial, conducted in the same health district, can be considered the most comparable as a similar LNS type supplement was used [15]. Huybregts and colleagues reported no meaningful effect on SGA, neither on PTB nor a list of anthropometric measures at birth. However, the intervention led to a positive effect on birth length (+0.6 to 6.7 mm). An important difference to take into account for comparison of the results is the use of MMN (MISAME-II) versus IFA tablets (MISAME-III) in the control group. Similarly, compared to IFA, prenatal LNS in The Gambia was not associated with SGA, birth weight, length, or head circumference [34]. Trials offering reduced amounts of LNS, so-called small-quantity LNS (20 g/d; 118 kcal/d), compared to IFA, found no effects on SGA prevalence, but reported increases in birth weight (3 to 166 g) and reduced risk of LBW (−4% to 61%) in Ghana [35], lower offspring stunting (−3% to 29%) in Bangladesh [36], and higher newborn MUAC (0.1 to 0.3 mm) in Malawi [37].

The observed effect of BEP on birth weight, with an increase of 8 to 92 g (95% CI), is comparable to earlier studies, reporting increments of 30 to 117 g [10] and 5 to 77 g [11]. This effect on birth weight and the 1 to 40 mm effect in birth length observed by our study can, at least partially, be attributed to the concurrent 0.4 to 2.5 days increment in GA at birth. Also, we speculate that the modest improvements in birth anthropometry is the result of the MMN compartment of the supplement, as previous research has shown that MMN led to an increase in birth weight resulting in lower proportion of LBW and SGA births [16].

Our data suggest that there was no risk in providing BEP to women that were not underweight at early gestation. The BEP did not impact the prevalence of LGA and no increase in C-sections was observed.

Subgroup analyses revealed that the intervention was efficacious in reducing SGA prevalence among mothers with a more adequate baseline nutritional status (e.g., nonanemic, higher MUAC). Similarly, in The Gambia, subgroup analysis indicated that the efficacy of BEP and/or MMN supplements might potentially be mediated through larger gestational weight gain (i.e., well-nourished women) [34]. These results are in contrast with previous findings showing that nutritional supplementation had larger treatment effects among inadequately nourished pregnant women at early gestation, including underweight mothers [15], women with negative energy balance [38], food insecure households [36], and primiparous mothers [35]. Furthermore, our subgroup analysis showed that the impact was more profound among female newborns, while other studies found a larger effect of nutritional supplementation in males [39,40].

Some explanations for the lack of strong efficacy of fortified BEP as compared to IFA can be put forward. First, frequent acute and chronic infections during pregnancy, which are often prevalent in LMICs [41], can lead to nutrient losses and nutrient sequestration in the mother, which, in turn, may have limited the quantity of nutrients available to the fetus. Our trial did not collect data on maternal infection during gestation, but if prevalent in this setting, acute or chronic infection may have reduced the efficacy of the BEP supplements provided. Likewise, acute or chronic infection in the child could have limited the potential benefits from the nutrients received by the fetus during pregnancy. Second, starting fortified BEP supplementation during early pregnancy alone might not be sufficient to prevent adverse birth outcomes. Our subgroup analyses indicated that the BEP intervention was potentially more efficacious among women who started pregnancy with a better nutritional status; hence, preconception supplementation may confer greater benefits on birth outcomes. Although the Women First trial found that providing LNS and BEP at least 3 months prior to conception did not yield additional benefits on child linear growth at birth relative to starting BEP supplementation during gestation [12], compelling evidence remains scarce and supplementation during the preconception period may warrant further exploration.

A major strength of our study was the high acceptability of the fortified BEP supplement, evaluated in a 2-phase formative study [21,22], and strong emphasis on daily observed intake. The high adherence rate reported in this trial (approximately 90% for IFA in both study groups and approximately 83% for BEP in intervention group) ensures the reliability of our results, compared to those from studies that rely on maternal recall of adherence. Moreover, the daily observed supplementation reduced the possible risk of sharing the supplement with other household members and supported micronutrient adequacy following existing requirements. A cross-sectional dietary intake assessment showed that BEP did not displace energy and nutrient intake from the usual diet [42]. Hence, we can almost rule out a substitution effect that could have limited the efficacy of the BEP to support fetal growth and reduce SGA. Another strength was the early enrollment of participants, as a result of a monthly visiting schedule at home by trained village-based project workers, who received refresher trainings

and close supervision by the MISAME-III field team. Finally, in almost all cases, birth weight was measured almost immediately after birth.

Our study also had some limitations. First, it was impossible to blind mothers or MISAME-III collaborators to the intervention allocation. Although care was taken to blind the study midwives measuring birth anthropometry, we cannot rule out that intervention allocation was unveiled by asking the mother which supplement she received. Second, it is possible that improvements are not visible through birth anthropometry and maternal biomarkers (to demonstrate any micronutrient deficiencies) or placental indicators are needed to assess an intermediate effect of the fortified BEP supplement on maternal nutritional status and placental function (e.g., fetal hypoxia might inhibit fetal growth) [15,43]. Ongoing multiomics substudies will provide insight into the biochemical profiles of mother infant dyads to address this current limitation. Third, we lacked data on maternal infection, inflammation, stress, and physical activity levels and could not determine the extent to which these prognostic risk factors may have influenced nutrient availability or poor birth outcomes [44].

In conclusion, we did not observe a statistically significant effect of fortified BEP supplements and IFA tablets on SGA prevalence, as compared to IFA tablets alone in rural Burkina Faso, although small positive effects were noticed on birth weight, GA, and LBW prevalence. Exploratory analyses suggests that prenatal BEP supplementation was more beneficial for mothers that enter pregnancy more adequately nourished. MISAME-III substudies will evaluate the efficacy of prenatal BEP and IFA tablets on additional maternal and child biochemical parameters to provide more insight in mechanisms of action and the clinical relevance of providing BEP supplementation.

## Supporting information

**S1 CONSORT checklist. Checklist of information to include when reporting a cluster randomized trial.** CONSORT, Consolidated Standards of Reporting Trials.
(DOCX)

**S1 Table. Complete cases analysis of primary and secondary outcomes.**
(DOCX)

**S2 Table. Per-protocol analyses of primary and secondary outcomes.**
(DOCX)

**S3 Table. Subgroup analysis by potential treatment effect modifiers of SGA.** SGA, small-for-gestational age.
(DOCX)

**S1 Fig. Treatment efficacy on birth weight across the distribution of maternal BMI.** The estimated difference in birth weight between the women who received the BEP supplement and IFA (intervention) and those who received only iron and folic acid (control) is shown as a function of the percentiles of maternal BMI. The zero line indicates no efficacy of BEP. The positive y values indicate a higher birth weight in the intervention group, and the negative y values indicate a lower birth weight. The central solid black line represents the smoothed treatment efficacy, with upper and lower dashed 95% confidence bands, using complete cases. BEP, balanced energy–protein; BMI, body mass index; IFA, iron–folic acid.
(TIF)

**S2 Fig. Treatment efficacy on birth length across the distribution of maternal BMI.** The estimated difference in birth length between the women who received the BEP supplement and IFA (intervention) and those who received only iron and folic acid (control) is shown as a

function of the percentiles of maternal BMI. The zero line indicates no efficacy of BEP. The positive y values indicate a higher birth length in the intervention group, and the negative y values indicate a lower birth length. The central solid black line represents the smoothed treatment efficacy, with upper and lower dashed 95% confidence bands, using complete cases. BEP, balanced energy–protein; BMI, body mass index; IFA, iron–folic acid.
(TIF)

## Acknowledgments

The authors thank all the women from Boni, Dohoun, Karaba, Dougoumato II, Koumbia, and Kari who participated in the study; the data collection team; and Henri Somé from AFRIC-Santé. We thank Nutriset (France) for donating the BEP supplements.

## Author Contributions

**Conceptualization:** Laeticia Celine Toe, Patrick Kolsteren, Lieven Huybregts, Carl Lachat.

**Data curation:** Brenda de Kok, Giles Hanley-Cook, Alemayehu Argaw, Katrien Vanslambrouck, Lieven Huybregts.

**Formal analysis:** Brenda de Kok, Giles Hanley-Cook, Alemayehu Argaw, Lieven Huybregts.

**Funding acquisition:** Patrick Kolsteren, Carl Lachat.

**Investigation:** Brenda de Kok, Laeticia Celine Toe, Giles Hanley-Cook, Moctar Ouédraogo, Anderson Compaoré, Katrien Vanslambrouck, Trenton Dailey-Chwalibóg.

**Methodology:** Brenda de Kok, Laeticia Celine Toe, Moctar Ouédraogo, Katrien Vanslambrouck, Patrick Kolsteren, Lieven Huybregts.

**Project administration:** Brenda de Kok, Laeticia Celine Toe, Moctar Ouédraogo, Katrien Vanslambrouck, Rasmané Ganaba, Patrick Kolsteren, Carl Lachat.

**Resources:** Brenda de Kok, Laeticia Celine Toe, Moctar Ouédraogo, Katrien Vanslambrouck.

**Software:** Brenda de Kok, Giles Hanley-Cook, Katrien Vanslambrouck, Lieven Huybregts.

**Supervision:** Patrick Kolsteren, Lieven Huybregts, Carl Lachat.

**Validation:** Laeticia Celine Toe, Patrick Kolsteren, Lieven Huybregts, Carl Lachat.

**Visualization:** Brenda de Kok, Giles Hanley-Cook.

**Writing – original draft:** Brenda de Kok, Laeticia Celine Toe, Giles Hanley-Cook.

**Writing – review & editing:** Brenda de Kok, Laeticia Celine Toe, Giles Hanley-Cook, Alemayehu Argaw, Trenton Dailey-Chwalibóg, Patrick Kolsteren, Lieven Huybregts, Carl Lachat.

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
