## [Editor Report · Decision Letter 0]

25 Feb 2022

Dear Dr Lachat, 

Thank you for submitting your manuscript entitled "Prenatal fortified balanced energy-protein supplementation and birth outcomes: a randomised controlled efficacy trial in rural Burkina Faso" for consideration by PLOS Medicine.

Your manuscript has now been evaluated by the PLOS Medicine editorial staff and I am writing to let you know that we would like to send your submission out for external peer review.

Please re-submit your manuscript within two working days, i.e. by Mar 01 2022 11:59PM.

Kind regards,

Beryne Odeny

PLOS Medicine

---

## [Decision Letter · Decision Letter 1]

22 Mar 2022

Dear Dr. Lachat,

Thank you very much for submitting your manuscript "Prenatal fortified balanced energy-protein supplementation and birth outcomes: a randomised controlled efficacy trial in rural Burkina Faso" (PMEDICINE-D-22-00628R1) for consideration at PLOS Medicine. 

[LINK]

In light of these reviews, I am afraid that we will not be able to accept the manuscript for publication in the journal in its current form, but we would like to consider a revised version that addresses the reviewers' and editors' comments. Obviously we cannot make any decision about publication until we have seen the revised manuscript and your response, and we plan to seek re-review by one or more of the reviewers. 

We expect to receive your revised manuscript by Apr 12 2022 11:59PM. Please email us (plosmedicine@plos.org) if you have any questions or concerns.

We look forward to receiving your revised manuscript. 

Sincerely,

Beryne Odeny, 

PLOS Medicine

plosmedicine.org

1) The title needs revision. Please remove the country from the subtitle and place it right before the colon, i.e, “… in rural Burkina Faso: A randomized controlled efficacy trial”

2) Data availability statement requires revision - If the de-identified participant data are not freely available, please describe briefly the ethical, legal, or contractual restriction that prevents you from sharing it. Please also include an appropriate contact (web or email address) for inquiries (this cannot be a study author).

3) Abstract:

a) Please provide the follow-up time and dates of the study

b) Please provide the number of participants lost to follow up in each arm

c) Please include a summary of adverse events if these were assessed in the study.

d) Please remove “trial registration” after abstract conclusion and include it the abstract “methods and findings”

4) Thank you for providing your CONSORT checklist. Please replace the page numbers with paragraph numbers per section (e.g. "Methods, paragraph 1"), since the page numbers of the final published paper may be different from the page numbers in the current manuscript.

5) Please label the CONSORT checklist as S1 Checklist.

6) Please move the Ethical statement to the Methods section.

7) Please replace the term “compliance” with “adherence” where it is used to refer to treatment adherence.

8) Replace “fetus” rather than “foetus

9) Figure 1 should be labeled “Fig 1. CONSORT flowchart” or similar

10) In the tables, please use “p value” instead of “P”

11) In the tables, please define symbol ∆

12) Please remove the ‘Funding” and “Conflict of interest”, from the end of the main text. In the event of publication, this information will be published as metadata based on your responses to the submission form

13) References

a) Please ensure journal name abbreviations consistently match those found in the National Center for Biotechnology Information (NCBI) databases

b) Please provide access dates for references with web links.

c) Ref #37 is incomplete – please include the journal name

Comments from the reviewers:

Reviewer #1: This manuscript is a extremely well-conducted study of maternal supplemental in a vulnerable population in the Sahel. The writing and documentation are coherent, thorough and transparent.

The daily observed administration of the supplement leaves no doubt that the findings genuinely reflect what occurs when the supplement is consumed.

My major ask is reconsider reporting other important outcomes in another manuscript. Make this a complete, coherent story about all of the effects of BEP supplementation in pregnancy. Tell us what BEP does to maternal weight gain and quality of life, birth outcomes and postnatal outcomes in a single story. Allow readers to quickly understand the full impact of the BEP supplement.

Minor additions to strengthen the manuscript would be to include a description of the context of the participants with regard to common infections. Is this area malaria endemic? What have other studies found with regard to sexually transmitted diseases? How much HIV is seen in the study context? Are there other endemic infectious conditions?

Was there any seasonal effect seen? Is there a season with more food insecurity and did the BEP supplement given more benefits during these times?

Reviewer #2: See attachment

Michael Dewey

Reviewer #3: This is an important study on a public health intervention, where despite global guidance, there is paucity of information from well designed population based studies. This is one of the well controlled and designed studies with early enough introduction the BEP, assessment of compliance and robust statistical analysis. The field methods had strong quality assurance methodology and the results therefore provide strong evidence that adds to our understanding of maternal nutrition interventions. 

The paper could be strengthened by attention to a few issues in the description of the design and analysis part of the study

1. Why was a 7% point reduction in SGA births chosen for sample size and power estimation? A lower figure could well have yielded a larger sample size and relevant impact

2. The authors focused on SGA as the primary outcome and found significant impact on preterm births. Given the excess risk of mortality and other health and growth outcomes in early infancy among newborn infants with are both preterm and SGA, why wasn't the composite outcome assessed as an outcome, even if secondary? 

3. The authors state that there was no displacement of food by the BEP supplement but do not present any intake data. If there are dietary intake data, these should be presented

[LINK]

---

## [Decision Letter · Decision Letter 2]

20 Apr 2022

Dear Dr. Lachat,

Thank you very much for re-submitting your manuscript "Prenatal fortified balanced energy-protein supplementation and birth outcomes in rural Burkina Faso: A randomised controlled efficacy trial" (PMEDICINE-D-22-00628R2) for review by PLOS Medicine.

I have discussed the paper with my colleagues and the academic editor and it was also seen again by two reviewers. I am pleased to say that provided the remaining editorial and production issues are dealt with we are planning to accept the paper for publication in the journal.

[LINK]

We look forward to receiving the revised manuscript by Apr 27 2022 11:59PM.   

Sincerely,

Beryne Odeny, 

PLOS Medicine

plosmedicine.org

Requests from Editors:

1) Please define the abbreviations in Tables and Figures e.g., CI, SD

2) Please remove the “Data sharing” statement at the end of the main text

Comments from Reviewers:

Reviewer #1: I have no comments, but am disappointed the authors will not include a report of the full dataset.

Reviewer #2: The authors have addressed all my points. I still do not like the +- notation but it is not worth fighting over.

Michael Dewey

[LINK]

---

## [Editor Report · Decision Letter 3]

27 Apr 2022

Dear Dr Lachat, 

On behalf of my colleagues and the Academic Editor, Dr. Zulfiqar A. Bhutta, I am pleased to inform you that we have agreed to publish your manuscript "Prenatal fortified balanced energy-protein supplementation and birth outcomes in rural Burkina Faso: A randomised controlled efficacy trial" (PMEDICINE-D-22-00628R3) in PLOS Medicine.

PRESS

Sincerely, 

Beryne Odeny 

PLOS Medicine